# Systematic Evaluation and Optimization of Unmanned Aerial Vehicle Tilt Photogrammetry Based on Analytic Hierarchy Process

**Zan Zhu [1,2], Jianqi Wang [1,2,*], Yating Zhu [3], Qiran Chen [1] and Xinyu Liang [4]**

[1] School of Aeronautics and Astronautics, Guilin University of Aerospace Technology, Guilin 541004, China; zanzhu@guat.edu.cn (Z.Z.); 2017040060447@guat.edu.cn (Q.C.)
[2] Guangxi Key Laboratory of UAV Telemetry, Guilin 541004, China
[3] School of Resources and Environment, Southwest University, Chongqing 400715, China; 112022320001469@swu.edu.cn
[4] School of Electronic Engineering and Automation, Guilin University of Electronic Science and Technology, Guilin 541000, China; 21082304064@mails.guet.edu.cn
* Correspondence: wangjianqi@guat.edu.cn; Tel.: +86-183-1436-9262

**Abstract:** In the data acquisition and processing of Unmanned Aerial Vehicle(UAV) oblique photography monomer modeling, it is hard to balance the model quality and the production efficiency. This research applied the Analytic Hierarchy Process (AHP) evaluation method in the field of systems engineering to the management and decision of UAV oblique photography monomer modeling. Firstly, the AHP model is constructed by the expert survey method, and the relative weights of 6 evaluation indicators in the first hierarchies and the comprehensive ones of 3 in the second hierarchies are calculated. Then, each index data of different photo modeling schemes were collected and processed through experiments, and the AHP model was used to systematically evaluate each modeling scheme. Finally, the quadratic function is constructed with two variables, the number of photos and the AHP system score, meanwhile obtaining the optimal scheme by calculation. This is a useful attempt to apply the scientific evaluation method in the field of systems engineering to the production management of UAV aerial surveys. In this way, the efficiency of internal and external data collection and processing can be maximally improved while guaranteeing modeling accuracy.

**Keywords:** analytic hierarchy process; tilt photogrammetry; system evaluation; quality control

## 1. Introduction

Nowadays, UAV platforms are a valuable source of data for mapping, surveillance, and 3D modeling [1]. At present, the research on UAV tilt photogrammetry mainly includes flight motion control and image data processing [2]. In the context of smart city construction and booming development, high precision, high-efficiency, and realistic 3D city models are some of the most important basic data. UAV tilt photogrammetry, as one of the most important technologies for conducting live-view 3D model production at present, has several advantages: it is highly efficient, economical favorable, and saves human and material resources. It has been widely used in the fields of map mapping [3], geological survey [4], disaster detection [5], environmental protection [6], engineering construction [7,8], agriculture and forestry [9], and digital city construction [10]. UAV tilt photography monolithic modeling can represent the detailed features of an object to be measured more finely than tilt photogrammetry modeling with five fixed lenses and routes. However, its disadvantage is also more obvious that the relatively low efficiency of internal and external data collection and processing. Therefore, how to improve the efficiency of internal and external data collection and processing, on the basis of ensuring 3D model quality? This is an important issue in the production management of UAV oblique photography monomer modeling. The production efficiency of UAV tilt photography monomer

modeling and the control and management of 3D model quality involve the influence of various elements. Therefore, building a control and management evaluation system for production efficiency and model quality is a complex system engineering task.

Analytic hierarchy process (AHP) is a system evaluation method for optimizing and evaluating systems with diverse object attributes and complex structures which are difficult to analyze completely quantitatively [11]. The method was first proposed by T.L. Saaty, an American operations researcher and professor at the University of Pittsburgh, and was used in the study of the U.S. Department of Defense's "Contingency Plan", which was introduced to the public at the First International Conference on Mathematical Modeling in 1977. This method is flexible, systematic, and concise, and has been widely used in economic management [12], public management and decision making [13–15], production management and decision making [16,17], environmental disasters and assessment [18], urban planning and landscape ecology [19,20], resource and environmental assessment and optimal allocation [21–23], education management [24,25], and other related research areas [26,27]. In summary, the analytic hierarchy process has had good results in the evaluation of highly complex structural systems, which are scientific methods for analysis, management, and decision making.

The management system of internal and external production and quality control for UAV oblique photography monomer modeling, involving multiple factors, is a complex system engineering, to which the evaluation and analysis need qualitative, quantitative as well as other systematic evaluation models and methods. However, no scholars have yet conducted a scientific and systematic analysis of the internal and external production and quality control system of UAV tilt photography monomer modeling using the AHP method in either management science or the field of UAV application technology. In this paper, the AHP system evaluation method is used as the theoretical basis to explore and analyze each system evaluation element and the relationship between them through experiments and to make decisions on the optimal modeling scheme so as to achieve system optimization.

## 2. AHP Method and System Evaluation Model Construction

### 2.1. AHP System Evaluation Method

The implementation of the AHP system evaluation method can be divided into four main steps: ① analyze and evaluate the relationship between the elements to establish the recursive hierarchy of the system; ② compare the elements of each level of the criterion layer by two, construct a judgment matrix, and conduct a consistency test; ③ calculate the relative weights under the criterion according to the judgment matrix; ④ calculate the synthetic weights of the elements of each level of the system to the total system objectives and rank the existing solutions.

### 2.2. Construction of the System Evaluation Model

2.2.1. System Evaluation Hierarchy Model Construction

As shown in Figure 1, the system evaluation model constructed by the AHP method can be divided into three layers: the target layer, the quasi-side layer, and the scheme layer. Among them, the purpose layer is used to determine the ultimate purpose of the system evaluation; the criterion level is used to determine the evaluation indicators of the system evaluation, which can be divided into different levels; and the scheme level mainly describes the schemes of the system to be evaluated. This paper analyzes the technical points of each link of UAV tilt photography and three-dimensional modeling. The system evaluation model based on this is shown in Figure 1.

In this model, layer *A* makes it clear that the ultimate goal of this study is to find the optimal scheme; the criterion layer includes three primary indicators of layer *B* and six secondary indicators of layer *C*. Among them, there are three indicators in layer *B*. The first level indicator is *B1* (occupied memory), which includes two secondary indicators: *C1* (photo memory) and *C2* (model memory). The second primary indicator is *B2* (time used), which includes two secondary indicators *C3* (field work consumption time) and *C4*

(time consuming in-house work). The third primary indicator is *B3* (model dimensional accuracy), which includes two secondary indicators: *C5* (model dimensional accuracy) and *C6* (model distortion area). These indexes in the criterion layer are the basis for evaluating various schemes of UAV tilt photogrammetry. The last layer is the scheme layer of the evaluation model, which mainly includes six UAV tilt photogrammetry schemes designed in this study.

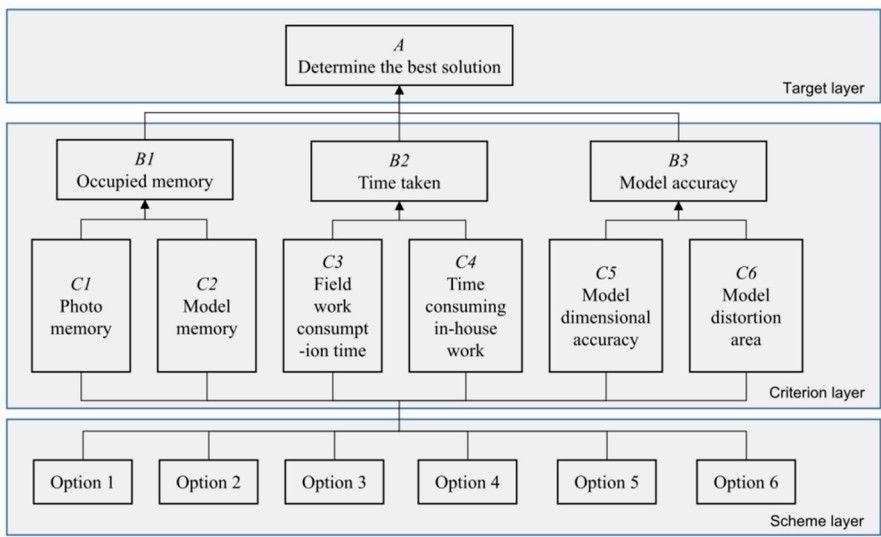

**Figure 1.** AHP structure model.

### 2.2.2. Construction of Judgment Matrix

In order to construct the judgment matrix, expert questionnaires were designed according to the scale shown in Table 1 and were sent to five units and enterprises of the Kunming University of Technology, the China University of Mining and Technology, the Guilin Institute of Aerospace Technology, the Yunnan Province Geology and Mapping Institute, and Guangxi Shengyao Aviation Technology (Nanning, China). The surveyed population all have rich technical backgrounds in production and research related to UAV tilt photography. Among them, 56.25% were technicians and engineers of production units, and 43.75% were teachers and researchers of universities.

**Table 1.** Scale definition of AHP judgment matrix.

| Scale | Meaning |
| --- | --- |
| 1 | Two elements of equal importance |
| 3 | The former is slightly more important than the latter |
| 5 | The former is obviously more important than the latter |
| 7 | The former is more important than the latter |
| 9 | The former is more important than the latter |
| 2, 4, 6, 8 | The middle value of the above two adjacent scales |
| reciprocal | Two elements, the latter is more important than the former (refer to the above scale for the degree) |

Through analysis of the expert questionnaire, the judgment matrices $U_{BA}$, $U_{CB1}$, $U_{CB2}$, and $U_{CB3}$ for the two levels of evaluation indicators $B_i$ and $C_i$ at the criterion level could be obtained as shown in Equations (1)–(4), respectively:

$$U_{BA} = \begin{bmatrix} b_{11} & b_{12} & b_{13} \\ b_{21} & b_{22} & b_{23} \\ b_{31} & b_{32} & b_{33} \end{bmatrix} = \begin{bmatrix} 1 & 1/3 & 1/5 \\ 3 & 1 & 1/4 \\ 5 & 4 & 1 \end{bmatrix} \quad (1)$$

$$U_{CB1} = \begin{bmatrix} C_{11} & C_{12} \\ C_{21} & C_{22} \end{bmatrix} = \begin{bmatrix} 1 & 2 \\ 1/2 & 1 \end{bmatrix} \tag{2}$$

$$U_{CB2} = \begin{bmatrix} C_{33} & C_{34} \\ C_{43} & C_{44} \end{bmatrix} = \begin{bmatrix} 1 & 1/4 \\ 4 & 1 \end{bmatrix} \tag{3}$$

$$U_{CB3} = \begin{bmatrix} C_{55} & C_{56} \\ C_{65} & C_{66} \end{bmatrix} = \begin{bmatrix} 1 & 1 \\ 1 & 1 \end{bmatrix} \tag{4}$$

### 2.2.3. Consistency Test

In order to ensure the reasonableness of the distribution of scores in the judgment matrix, a consistency test was required. The basic method is as follows: Firstly, the consistency index ($C.I.$) is calculated according to Equation (5), then the random consistency index ($R.I.$) is calculated by checking Table 2. Finally, the consistency ratio ($C.R.$) is calculated according to Equation (6) and the consistency test of the judgment matrix is performed based on the consistency ratio.

$$C.I. = \frac{\lambda - n}{n - 1} \tag{5}$$

where $\lambda$ is the maximum characteristic root of the judgment matrix and $n$ is the order of the judgment matrix.

$$C.R. = \frac{C.R.}{R.I.} \tag{6}$$

**Table 2.** Average random consistency index.

| Matrix Order ($n$) | 1 | 2 | 3 | 4 | 5 | 6 | 7 | 8 | 9 | 10 |
|---|---|---|---|---|---|---|---|---|---|---|
| Random consistency index ($R.I.$) | 0 | 0 | 0.52 | 0.89 | 1.12 | 1.26 | 1.36 | 1.41 | 1.46 | 1.49 |

Analysis of the four matrices constructed in this system show that the matrix $U_{BA}$ is of order three, and $U_{CB1}$, $U_{CB2}$ and $U_{CB3}$ are all of order two. Therefore, only the matrix $U_{BA}$ needs to be tested for consistency. The maximum characteristic root $\lambda$ of the judgment matrix $U_{BA}$ is 3.086 and the consistency index ($C.I.$) can be calculated as 0.043 by substitution into Equation (5). The random consistency index ($R.I.$) of the matrix can be obtained as 0.52 by checking Table 2. The consistency ratio ($C.R.$) of the judgment matrix $U_{BA}$ can be obtained as 0.083 < 0.1 by substituting the above two indicators into Equation (5). It was determined that the constructed judgment matrix $U_{BA}$ passes the consistency test and can be used in subsequent calculations.

### 2.2.4. Calculation of Relative Weights and Synthetic Weights

(1)    Calculation of relative weights

Based on the four judgment matrices of $U_{BA}$, $U_{CB1}$, $U_{CB2}$, and $U_{CB3}$ constructed above, the relative weights of each element in $B$ and $C$ relative to the previous layer can be calculated by Equations (7) and (8), respectively.

$$W_i = \left( \prod_{j=1}^{n} a_{ij} \right)^{\frac{1}{n}} \tag{7}$$

$$W_i^0 = \frac{W_i}{\sum\limits_i W_i} \tag{8}$$

The results of the calculation of the relative weights are shown in Table 3:

**Table 3.** Relative weight calculation results.

| B Layer | | | C Layer | | |
|---|---|---|---|---|---|
| **Evaluating Indicator** | $W_i$ | $W_{i0}$ | **Evaluating Indicator** | $W_i$ | $W_{i0}$ |
| B1 | 0.405 | 0.101 | C1 | 1.414 | 0.667 |
| | | | C2 | 0.707 | 0.333 |
| B2 | 0.909 | 0.226 | C3 | 0.500 | 0.200 |
| | | | C4 | 2.000 | 0.800 |
| B3 | 2.714 | 0.674 | C5 | 1.000 | 0.500 |
| | | | C6 | 1.000 | 0.500 |

(2)  Synthetic weight calculation

Based on the weights of each indicator in layer *B*, then calculate the comprehensive weight of each indicator in layer *C* in the system evaluation according to Equation (9), the calculation results are shown in Table 4.

$$C_j = \sum_{i=1}^{n} b_i C_j^i \tag{9}$$

where $b_i$ is the relative weight of each indicator in layer *B* and $C_{ij}$ is the relative weight of each indicator in layer *C* relative to layer *B*.

**Table 4.** Composite weight calculation.

| | **B1 (0.101)** | **B2 (0.226)** | **B3 (0.674)** | **Composite Weight** |
|---|---|---|---|---|
| C1 | 0.667 | 0 | 0 | 0.067 |
| C2 | 0.333 | 0 | 0 | 0.034 |
| C3 | 0 | 0.200 | 0 | 0.045 |
| C4 | 0 | 0.800 | 0 | 0.180 |
| C5 | 0 | 0 | 0.500 | 0.337 |
| C6 | 0 | 0 | 0.500 | 0.449 |

## 3. Experimental Process

### 3.1. Experimental Scheme Design

In the literature [28], the effects of 3D models of mountainous and snowy areas constructed by different types of UAV oblique photogrammetry are compared and analyzed, and the beneficial results are obtained, where the provided experimental schemes, image processing techniques and data analysis methods are important references for this paper. The objective of this experiment is to evaluate alternative solutions for the production of UAV tilt photography monomer modeling with different orders of magnitude of photos by the AHP method, and to select the optimal solution. The specific experiment includes the following steps:

(1)  Scheme design. Six shooting scenarios for tilt photography modeling were set up with different orders of magnitude of photos. The number of photos for the six shooting scenarios are 10, 30, 50, 70, 90 and 110.

(2)  External data collection. The UAV was manipulated to obtain external data, meanwhile recording the time spent on external photography shooting of each scheme (indicator *C3*).

(3)  Internal processing. Internal data processing was conducted, including air three encryption and 3D modeling processing. The photo memory (index *C1*), model memory (index *C2*), and internal processing time (index *C4*) were recorded separately in this process.

(4) Data verification. The constructed real-world 3D model was verified by measuring the building in the field and the model size accuracy (index *C5*) and model distortion area (index *C6*) were obtained.

(5) System evaluation calculation. The evaluation scores of each alternative were calculated based on the data of each indicator in C layer obtained from the experiment and the synthetic weights were constructed using the AHP method (as shown in Table 4).

(6) Program optimization. The number of photographs taken and the system score of the AHP method were used as the two variables to construct polynomials and the maximum number was calculated by integration, which was used as the optimal solution.

### 3.2. Tilt Photography Field Data Acquisition

#### 3.2.1. Research Objects and Equipment

According to the experimental needs, a single building (surface area approximately 1005.24 m$^2$ and building volume approximately 3069.87 m$^3$) was selected as the research object in this paper (Figure 2). The tilt photography monolithic modeling field data acquisition was carried out by DJI Genie Phantom 4 Pro V2.0 (SZ DJI Technology Co., Ltd., Shenzhen, China).

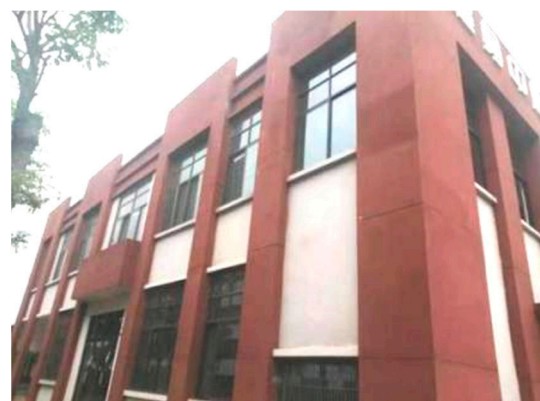

**Figure 2.** Research Area.

#### 3.2.2. Field Data Collection

The field data photography was done by the same experienced UAV aerial survey field operator and the data collection was completed for each of the six alternative scenarios according to the plan. The time taken to complete each option (indicator *C3*) was also recorded, indicator *C3* was evaluated according to the scale evaluation criteria shown in Table 5, and the obtained data and corresponding evaluation scores were obtained as shown in Table 6.

**Table 5.** Evaluation index scale table.

| Evaluation Interval | [0–20%] | (20–40%] | (40–60%] | (60–80%] | (80–100%] |
|---|---|---|---|---|---|
| Evaluation score | 5 | 4 | 3 | 2 | 1 |

### 3.3. Internal Data Processing and Validation

#### 3.3.1. 3D Model Construction

The three-dimensional model construction processing experiment was performed using the Context Capture software platform with the following computer configuration: Intel (R) Core(TM) i7-7700 CPU@3.60 GHz 1.99 GHz processor and 16 GB RAM. Processing mainly included empty three encryption and three-dimensional model construction two steps. Each program built a three-dimensional model, shown in Figure 3.

**Table 6.** Experimental data acquisition and scale evaluation of *C* layer.

| Option (Number of Photos) | Photo Memory *C1* (0.017) | | Model Memory *C2* (0.084) | | Field Shooting Time *C3* (0.113) | | Office Processing Time *C4* (0.113) | | Model Dimensional Accuracy *C5* (0.168) | | Model Distortion Area *C6* (0.505) | | Comprehensive Evaluation of the Scheme |
|---|---|---|---|---|---|---|---|---|---|---|---|---|---|
| | Raw Data/M | Scale Evaluation | Raw Data/M | Scale Evaluation | Raw Data/min | Scale Evaluation | Raw Data/min | Scale Evaluation | Raw Data/cm | Scale Evaluation | Raw Data/cm$^2$ | Scale Evaluation | |
| Option 1 (10) | 74.2 | 5 | 323.0 | 5 | 5.0 | 5 | 8.6 | 5 | 46.6 | 1 | 469.4 | 1 | 2.30 |
| Option 2 (30) | 222.0 | 5 | 1013.8 | 4 | 25.0 | 5 | 15.8 | 5 | 9.6 | 4 | 255.1 | 2 | 3.63 |
| Option 3 (50) | 374.0 | 2 | 1648.6 | 3 | 42.0 | 4 | 23.6 | 4 | 3.4 | 5 | 182.4 | 4 | 4.20 |
| Option 4 (70) | 523.0 | 2 | 2457.6 | 2 | 69.0 | 2 | 30.3 | 3 | 4.0 | 5 | 167.0 | 5 | 4.16 |
| Option 5 (90) | 671.0 | 1 | 2979.8 | 1 | 89.0 | 1 | 37.5 | 2 | 3.9 | 5 | 126.7 | 5 | 3.90 |
| Option 6 (110) | 822.0 | 1 | 3225.6 | 1 | 102.0 | 1 | 44.7 | 1 | 3.6 | 5 | 105.9 | 5 | 3.70 |

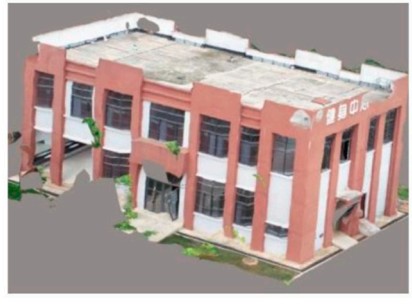
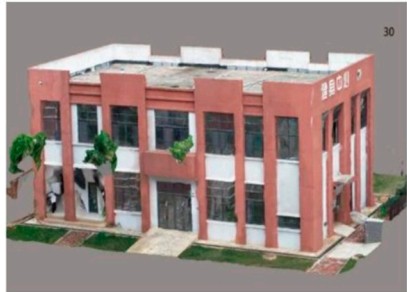
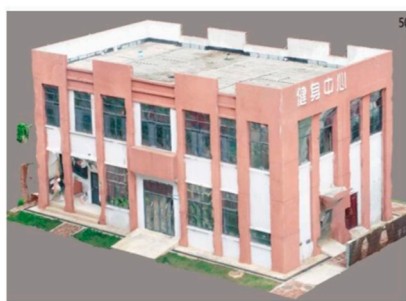

a.Option 1 (10 photos)　　　　　b.Option 2 (30 photos)　　　　　c.Option 3 (50 photos)

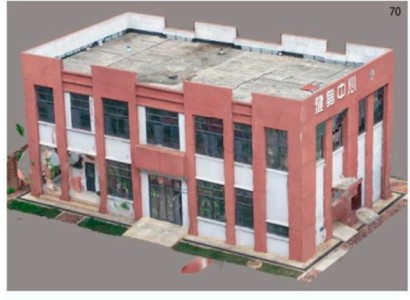
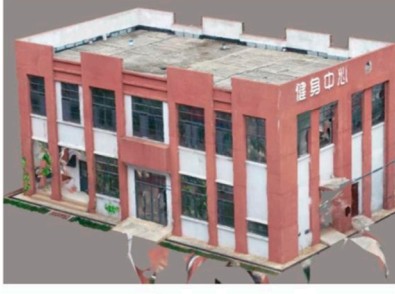
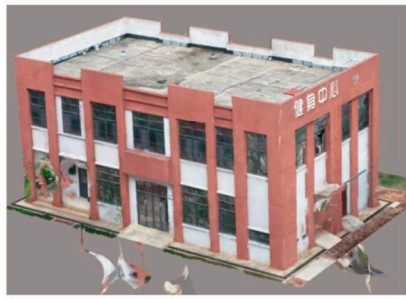

d.Option 4 (70 photos)　　　　　e.Option 5 (90 photos)　　　　　f.Option 6 (110 photos)

**Figure 3.** 3D model of each scheme. In this process, the next photo memory (indicator *C1*), model memory (indicator *C2*) and internal processing time (indicator *C4*) are recorded, the three indicators are evaluated according to the scale evaluation criteria shown in Table 5, and the obtained data and corresponding evaluation scores are shown in Table 6.

### 3.3.2. Data Validation

After constructing the real-world 3D model, the accuracy of the model was verified by field measurement, mainly including model size accuracy (index *C5*) and model distortion area (index *C6*). The three indexes were evaluated according to the scale evaluation criteria shown in Table 5. The obtained data and corresponding evaluation scores are shown in Table 6.

## 4. Systematic Evaluation and Optimal Design of Shooting Solution

### 4.1. AHP Method System Evaluation Calculation and Analysis

Based on the synthetic weights calculated by the AHP evaluation method, the six shooting scenarios were evaluated and calculated by Equation (10). The evaluation results are shown in Table 5.

$$V_i = \sum_{j=1}^{n} V_{ij} C_j \tag{10}$$

where $C_j$ is the comprehensive weight corresponding to the evaluation index *j* calculated by the AHP method, $V_i$ is the comprehensive evaluation score of scheme *i*, and $V_{ij}$ is the scale evaluation of index *j* corresponding to scheme *i*.

Through comparison of the system evaluation results, it can be concluded that option three (50 photos) is optimal.

### 4.2. Scheme Optimization and Verification

#### 4.2.1. Mathematical Model Selection

Taking [29] as a reference, this study uses the method of a mathematical function fitting to build a mathematical model and evaluates the accuracy of each scheme based on this mathematical model. Through the statistical analysis of the above experimental data, we can choose the number of photos and AHP system score as two variables and take

the number of photos as variable $x$ and AHP system score as variable $y$. Through analysis of the relationship between these two variables with the numerical analysis method, a mathematical model can be constructed to simulate the relationship between variable $x$ and variable $y$. Generally, there are two kinds of methods to build mathematical models between two variables: the linear function method and the polynomial fitting method. The linear function constructed for these two variables is shown in Figure 4a, and the quadratic term function constructed is shown in Figure 4b. The closer the value of $R^2$ in the figure is to 1, the higher the reliability of the function model.

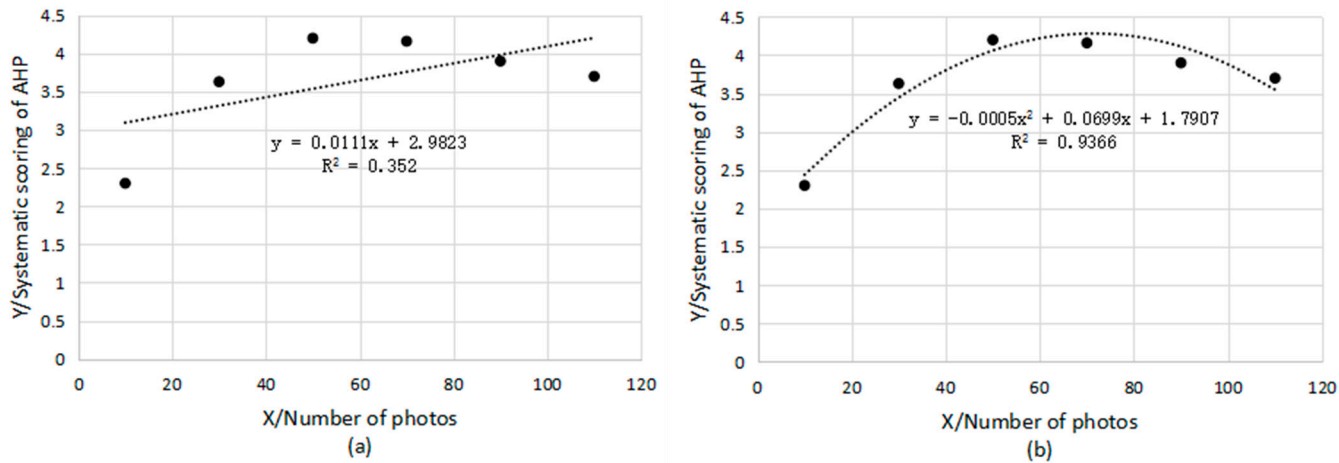

**Figure 4.** (**a**) Linear fitting function model. (**b**) Quadratic fitting function model.

The linear fitting function model constructed, as shown in Figure 4a, is shown in Equation (11). The quadratic fitting function model constructed as shown in Figure 4b is shown in Equation (12). In order to evaluate the advantages and disadvantages of the two function models shown in Figure 4, this study adopts the mathematical model accuracy evaluation method provided by references [29,30]. In this method, *MSE* (mean square error), *RMSE* (root mean square error), and $R^2$ (coefficient of determination) are used as the basis for evaluation of the reliability of the functional model. The statistical object of these three indicators is the deviation between the predicted value $y'$ and the true value $y$ of the function model $\Delta y$. The calculation process is carried out according to Equations (14)–(16).

$$y = 0.0111x + 2.9823 \tag{11}$$

$$y = -0.0005x^2 + 0.0698x + 1.7931 \tag{12}$$

$$\Delta y = y' - y \tag{13}$$

$$MSE = \frac{\sum\limits_{i=1}^{n} (y_i' - y_i)^2}{n} \tag{14}$$

$$RMSE = \sqrt{\frac{\sum\limits_{i=1}^{n} (y_i' - y_i)^2}{n}} \tag{15}$$

$$R^2 = 1 - \frac{\sum_i (y_i' - y_i)^2}{\sum_i (y_i - \overline{y})} \tag{16}$$

The deviation value is counted according to the above formulae, and the comparison of statistical results is shown in Table 7.

**Table 7.** Statistics of deviation value of function model.

| Number of Photos/$x$ | Truth Value/$y'$ | Linear Fitting Function Model | | | | | Quadratic Fitting Function Model | | | | |
|---|---|---|---|---|---|---|---|---|---|---|---|
| | | Estimate/$y$ | Deviation Value/$\Delta y$ | *MSE* | *RMSE* | $R^2$ | Estimate/$y$ | Deviation Value/$\Delta y$ | *MSE* | *RMSE* | $R^2$ |
| 10 | 2.3 | 3.09 | 0.79 | | | | 2.43 | 0.14 | | | |
| 30 | 3.63 | 3.32 | −0.31 | | | | 3.44 | −0.19 | | | |
| 50 | 4.2 | 3.54 | −0.66 | 0.26 | 0.51 | 0.35 | 4.04 | −0.16 | 0.03 | 0.17 | 0.94 |
| 70 | 4.16 | 3.76 | −0.40 | | | | 4.23 | 0.07 | | | |
| 90 | 3.9 | 3.98 | 0.08 | | | | 4.03 | 0.13 | | | |
| 110 | 3.7 | 4.20 | 0.50 | | | | 3.43 | −0.27 | | | |

### 4.2.2. Scheme Optimization Analysis

As shown in Figure 4b, the linear relationship between the number of photos taken by tilt photography monomer modeling and the system score based on the AHP method can be constructed, and the corresponding quadratic term function can be fitted, as shown in Equation (17).

$$y = -0.0005x^2 + 0.0699x + 1.7931 \tag{17}$$

The derivation yields:

$$y' = -0.001x + 0.0699 \tag{18}$$

The maximum value of this function was calculated to be 70, which leads to the conclusion that the highest evaluation score was obtained when the number of photos taken was 70 for this building.

In a follow-up study, the system evaluation model and optimization method proposed in this paper were used in five different UAV tilt photogrammetry projects of two companies, and the overall production efficiency increased by 72.4%.

## 5. Discussion

This paper uses AHP as its core method and completes the following work of scheme optimization of the entire working process of UAV tilt photogrammetry and obtains useful results.

(1) Through the expert investigation method, the analytic hierarchy process model for the evaluation of tilt photogrammetry monomer modeling scheme was constructed and the comprehensive weight of the evaluation index was calculated.
(2) The analytic hierarchy process model was substituted into the specific experiment for calculation and evaluation analysis.
(3) The optimal shooting scheme was calculated by building a mathematical model, and the reliability of this method was verified by an experimental case.
(4) The system evaluation and scheme optimization method proposed in this paper was applied to production practice, and the work efficiency was increased by 32.4%.

In this field, future research can add multi-dimensional and multi-level evaluation indicators, such as the complexity of building structure, building volume, surface area, and color, so as to build a more refined analytic hierarchy process model in order to realize systematic and scientific analyses, obtain control of the entire process of tilt photography 3D modeling, and finally, achieve scientific decision-making and optimization of the scheme.

**Author Contributions:** Data curation, Z.Z.; formal analysis, Z.Z.; funding acquisition, J.W.; resources, X.L.; software, Q.C.; supervision, J.W.; writing—original draft, Z.Z.; writing—review and editing, Y.Z. All authors have read and agreed to the published version of the manuscript.

**Funding:** This research was funded by The National Natural Science Foundation of China (NNSFC, 61966010).

**Institutional Review Board Statement:** Not applicable.

**Informed Consent Statement:** Not applicable.

**Data Availability Statement:** All the required data is included in the paper.

**Acknowledgments:** This study was financially supported by the National Natural Science Foundation of China (No. 61966010) and was supported by the equipment provided by the Key Laboratory of UAV Telemetry in Guangxi Universities in this study and by the technical support of several researchers and engineers from the Kunming University of Technology (Kunming, China), the China University of Mining and Technology (Xuzhou, China), the Guilin Institute of Aerospace Technology (Guilin, China), the Yunnan Geological and Mining Survey Institute (Kunming, China), and Guangxi Shengyao Aviation Technology Co., Ltd. (Nanning, China).

**Conflicts of Interest:** The funders had no role in the design of the study; in the collection, analyses, or interpretation of data; in the writing of the manuscript; or in the decision to publish the results.

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
