# Peer review of "Systematic Evaluation and Optimization of Unmanned Aerial Vehicle Tilt Photogrammetry Based on Analytic Hierarchy Process"

_applsci, doi:10.3390/app12157665_

Round 1
Reviewer 1 Report
The manuscript entitled "Systematic evaluation and optimization of UAV tilt photogrammetry scheme based on AHP" discusses an important subject.
Comments:
1- UAV and AHP must be defined in the abstract and introduction.
2- Single example does not enough to verify the proposed method. Another example is needed.
3- Figure 4 shows a non-linear relationship, it does not linear, please correct or explain your justification.
4- Compare the result with the literature to show the novelty
Author Response
Response to Reviewer 1 Comments
Point 1: 1)It is undesirable to use abbreviations in the title of the article
Response 1: I have modified the abbreviations "AHP" and "UAV" in the title of the article into complete nouns.The modified content has been marked in red in the document.
Point 2: There is no complete and correct decoding of AHP at the first mention.
Response 2: I have modified the "AHP" and "UAV" that appear for the first time in this article into a complete and correct decoding.The modified content has been marked in red in the document.
Point 3:The review should also be expanded with important results of the application of mathematical models for UAV control (10.3390/s22093399, 10.1007/978-3-030-29513-4_74, 10.5194/isprsarchives-XXXVIII-1-C22-25-2011), as well as computer of vision, which can also be used to build depth maps
Response 3: I have read these three literatures and have a supplementary discussion in the introduction.The modified content has been marked in red in the document.
Point 4:It is necessary to give a detailed description of Figure 1.
Response 4: I have described Figure 1 in more detail in section 2.2.2 of this article.The modified content has been marked in red in the document.
Point 5: list enumerations are best done with line wrapping
Response 5: I have modified this article according to your comments.The modified content has been marked in red in the document.
Point 6: 3.2.1 indexes 2 and 3 after meters should be super-index
Response 6: I have modified section 3.2.1 according to your comments.The modified content has been marked in red in the document.
Point 7: Table 2 is too hard to read. Maybe it's worth splitting the information into 2 tables.
Response 7: I have revised table 2 according to your comments.The modified content has been marked in red in the document.
Point 8: Fig. 4. What is the reason for the choice of a quadratic approximation, why are other functions not checked?
Response 8: According to your opinion, I modified Figure 4, added the specific analysis process for linear fitting and polynomial fitting, and explained why the polynomial fitting model was selected as the system optimization analysis model.The modified content has been marked in red in the document.
Point 9: The conclusions must reflect the quantitative assessments of the quality that are obtained as a result of the study.
Response 9: According to your opinion, I have modified the experimental analysis and conclusion of the article and added the content of quantitative analysis of the experimental results.The modified content has been marked in red in the document.
Please see the attachment!
Response to Reviewer 2 Comments
Point 1: UAV and AHP must be defined in the abstract and introduction.
Response 1: Based on your comments, I have fully defined UAV and AHP in the summary and introduction.The modified content has been marked in red in the document.
Point 2: Single example does not enough to verify the proposed method. Another example is needed.
Response 2: According to your opinion, I have counted the efficiency improvement effect of this method in actual production, obtained useful results, and modified it in the article.The modified content has been marked in red in the document.
Point 3: Figure 4 shows a non-linear relationship, it does not linear, please correct or explain your justification.
Response 3: I have modified Figure 4 according to your opinion and added the content of polynomial fitting and linear fitting comparison.The modified content has been marked in red in the document.
Point 4: Compare the result with the literature to show the novelty
Response 4: According to your opinion, I have added the research content of comparative analysis to the article.The modified content has been marked in red in the document.
Please see the attachment!

Reviewer 2 Report
The article is devoted to the optimization of the control of an unmanned aerial vehicle through the systematic evaluation of the photogrammetry map. The authors obtained an interesting result, but the article needs a number of improvements:
1) It is undesirable to use abbreviations in the title of the article
2) There is no complete and correct decoding of AHP at the first mention.
3) The review should also be expanded with important results of the application of mathematical models for UAV control (10.3390/s22093399, 10.1007/978-3-030-29513-4_74, 10.5194/isprsarchives-XXXVIII-1-C22-25-2011), as well as computer of vision, which can also be used to build depth maps
4) It is necessary to give a detailed description of Figure 1.
5) list enumerations are best done with line wrapping
6) 3.2.1 indexes 2 and 3 after meters should be super-index
7) Table 2 is too hard to read. Maybe it's worth splitting the information into 2 tables.
8) Fig. 4. What is the reason for the choice of a quadratic approximation, why are other functions not checked?
9) The conclusions must reflect the quantitative assessments of the quality that are obtained as a result of the study.
Author Response
Point 1: UAV and AHP must be defined in the abstract and introduction.
Response 1: Based on your comments, I have fully defined UAV and AHP in the summary and introduction.The modified content has been marked in red in the document.
Point 2: Single example does not enough to verify the proposed method. Another example is needed.
Response 2: According to your opinion, I have counted the efficiency improvement effect of this method in actual production, obtained useful results, and modified it in the article.The modified content has been marked in red in the document.
Point 3: Figure 4 shows a non-linear relationship, it does not linear, please correct or explain your justification.
Response 3: I have modified Figure 4 according to your opinion and added the content of polynomial fitting and linear fitting comparison.The modified content has been marked in red in the document.
Point 4: Compare the result with the literature to show the novelty
Response 4: According to your opinion, I have added the research content of comparative analysis to the article.The modified content has been marked in red in the document.
Please see the attachment!

Round 2
Reviewer 1 Report
No more corrections are required.
Author Response
Thank you very much for your review.
Reviewer 2 Report
To some extent, the comments have been corrected, but not completely. It is still strongly recommended that authors pay attention to management methods based on mathematical modeling (10.3390/s22093399, 10.1007/978-3-030-29513-4_74, computer vision, which can also be used to build depth maps (10.1109/ACCESS.2020.2971938, 10.1016/j.coldregions.2021.103344, 10.3390/sym14010148).
Author Response
Point 1: To some extent, the comments have been corrected, but not completely. It is still strongly recommended that authors pay attention to management methods based on mathematical modeling (10.3390/s22093399, 10.1007/978-3-030-29513-4_74, computer vision, which can also be used to build depth maps (10.1109/ACCESS.2020.2971938, 10.1016/j.coldregions.2021.103344, 10.3390/sym14010148).
Response 1: The main content of this paper is to apply AHP method to the quality evaluation and scheme optimization of three-dimensional modeling of UAV tilt photogrammetry. The research focuses on the evaluation and optimization of the whole process of modeling scheme, rather than the specific image processing algorithm. However, there are still defects in the specific analysis methods in this paper. According to your comments, I have referred to the literature (10.3390/s22093399, 10.1016/j.coldregions.2021.103344) to improve the experimental scheme. In addition, I also refer to the method of evaluating the reliability of function model fitting in the literature (10.1109/access.2020.2971938, 10.3390/sym14010148), and make a more in-depth analysis of the accuracy of the linear fitting model and the quadratic fitting model in this paper, so as to finally realize the quantitative evaluation of the analysis results. Thank you very much for your review.